

# Evidence-based economic analysis demonstrates that ecosystem service benefits of water hyacinth management greatly exceed research and control costs

Lisa A. Wainger[1], Nathan E. Harms[2], Cedric Magen[1], Dong Liang[1], Genevieve M. Nesslage[1], Anna M. McMurray[1,3] and Al F. Cofrancesco[2]

[1] Chesapeake Biological Laboratory, University of Maryland Center for Environmental Science, Solomons, MD, USA
[2] Engineer Research and Development Center, US Army Corps of Engineers, Vicksburg, MS, USA
[3] Current affiliation: Winrock International, Arlington, VA, USA

## ABSTRACT

Invasive species management can be a victim of its own success when decades of effective control cause memories of past harm to fade and raise questions of whether programs should continue. Economic analysis can be used to assess the efficiency of investing in invasive species control by comparing ecosystem service benefits to program costs, but only if appropriate data exist. We used a case study of water hyacinth (*Eichhornia crassipes* (Mart.) Solms), a nuisance floating aquatic plant, in Louisiana to demonstrate how comprehensive record-keeping supports economic analysis. Using long-term data sets, we developed empirical and spatio-temporal simulation models of intermediate complexity to project invasive species growth for control and no-control scenarios. For Louisiana, we estimated that peak plant cover would be 76% higher without the substantial growth rate suppression (84% reduction) that appeared due primarily to biological control agents. Our economic analysis revealed that combined biological and herbicide control programs, monitored over an unusually long time period (1975–2013), generated a benefit-cost ratio of about 34:1 derived from the relatively modest costs of $124 million ($2013) compared to the $4.2 billion ($2013) in benefits to anglers, waterfowl hunters, boating-dependent businesses, and water treatment facilities over the 38-year analysis period. This work adds to the literature by: (1) providing evidence of the effectiveness of water hyacinth biological control; (2) demonstrating use of parsimonious spatio-temporal models to estimate benefits of invasive species control; and (3) incorporating activity substitution into economic benefit transfer to avoid overstating benefits. Our study suggests that robust and cost-effective economic analysis is enabled by good record keeping and generalizable models that can demonstrate management effectiveness and promote social efficiency of invasive species control.

Corresponding author
Lisa A. Wainger,
wainger@umces.edu

## WHY CONDUCT ECONOMIC ANALYSIS OF INVASIVE SPECIES?

Robust demonstration of invasive species management program benefits may be crucial to maintaining programs with tight budgets over the long term, particularly if programs are successful at reducing harms, thereby reducing the apparent urgency of action. Yet such analyses are not possible if agencies do not collect the appropriate data and information. Economic analyses require robust cause-and-effect relationships be established between management actions and environmental changes in order to design control strategies to maximize net benefits (*Shackelford et al., 2013*) and avoid spending when success rates are low (*Wainger et al., 2010*). However, the necessary data are often incomplete, largely because conditions prior to control and the effectiveness of control are not fully documented or records are not maintained for long enough.

Control of harmful non-native invasive species is the type of management decision that can benefit from a thorough economic analysis comparing benefits and costs for several reasons. First, efficient spending is needed since there are always more threats to species and ecosystems than resources to confront them (*Wilcove et al., 1998*). Second, the ongoing debate about whether we are vilifying invasive species without cause, and thereby wasting resources on their control (*Lodge & Shrader-Frechette, 2003*; *Shackelford et al., 2013*), can often be resolved through economic analyses. While much of the economic literature examines potential optimal control, retrospective analyses of actions are useful because they reveal the harms averted through diligent management and the value of ongoing management or prevention.

## WHAT INFORMATION SUPPORTS A SOLID ECONOMIC ANALYSIS?

For economic evaluations to enable efficient allocation of scarce invasive control resources, values (or indicators of value) must measure outcomes for which people would be willing to pay or otherwise trade off other goods and services. This basic concept, which is fundamental to economic theory, is often obfuscated by studies that quantify benefits of invasive species control in monetary terms that are not measuring economic benefits. In particular, many studies have equated reduced costs of control with benefits (*Sinden & Griffith, 2007*). However, changes in costs of invasive species control do not directly demonstrate that the spending was in the public interest.

The economic measure of benefits, *utility*, encompasses all tangible and intangible effects on well-being, not only financial effects (*Freeman, Herriges & Kling, 2014*). As a result, a wide array of environmental changes will be relevant to cost benefit or cost-effectiveness analysis, if they can be connected to human concerns. For example, many studies have documented peoples' willingness to pay to retain rare species (*Richardson & Loomis, 2009*). These *nonuse* values represent intangible benefits derived from stewardship of the environment. For aquatic species, previously measured benefits include these nonuse values and use values from recreational and commercial fishing, energy supply,

water supply, agriculture, industry, tourism, property value support, and flood damages avoided (*Lovell, Stone & Fernandez, 2006*).

Even with the best data collection, cost-benefit analysis of proposed invasive species management requires modeling to fill in for unobservable data. A substantial challenge to analyzing benefits of past control is developing the *counterfactual* or without-action scenario that is needed to understand how management changes outcomes. Data-rich case studies reveal the methods that can be used to dynamically integrate socio-ecological systems and quantify benefits of invasive species control (as described in *Olson, 2006*). However, sophisticated models are time-consuming or impossible to build for many data-poor case studies and their use is far from routine for evaluating management options.

To ease some of the analytic burdens of measuring economic benefits, many government agencies use economic benefit transfer models for routinely estimating program benefits (*Wilson & Hoehn, 2006*). Economic benefit transfer is the process of transferring values that have been empirically estimated for one or more locations to unstudied, or transfer, locations (*Johnston et al., 2015*). Although this approach is subject to error, the accuracy is generally increased by using studies that are closely matched to the transfer site (*Plummer, 2009*) or using functional models to adjust values based on the social, economic, and ecological conditions of the site (*Johnston & Wainger, 2015*).

Even with benefit transfer, substantial work can be required to apply the technique to quantify how environmental changes impact human concerns (*Johnston & Wainger, 2015*; *Mazzotta et al., 2015*). Modeling complexity can be compounded when temporal and spatial detail is required to accurately assess impacts. For example, the degree of overlap in time between aquatic plant growth and seasonal boat-dependent activities can determine degree of harm (*Adams & Lee, 2007*). Similarly, a spatial framework may be needed to accurately model effects of proposed management, such as using barriers to prevent range expansion (*Sharov, 2004*; *Rahel & Olden, 2008*).

An often overlooked component of system response in benefit transfer is the degree to which people adapt to change, which if not considered, can inflate value estimates. For example, *Keller, Frang & Lodge (2008)* valued the benefits of preventing invasion of rusty crayfish into lakes as the *elimination* of spending by anglers seeking panfish, since these fish would be extirpated by the invasion. Yet it is plausible that at least some anglers would switch to alternative lakes or alternative species and enjoy comparable or modestly reduced benefits. Assuming all benefits are lost from an environmental change is common in benefit transfer because data for generalizing how people adapt are poor. Site-specific studies with detailed data can be used to estimate substitution of sites or other adaptations.

Although it is always tempting to add model complexity to better capture human-environment interactions and reduce error, complexity also tends to reduce the transferability of methods and findings and increases data requirements. In the analysis that follows, we sought to create parsimonious models to support routine economic analyses of invasive effects (see *Robinet et al., 2012*). Yet we also wanted to include relevant spatial, temporal and behavioral detail to reduce error of estimates. To achieve these goals, our modeling approach combined economic benefit transfer with moderately detailed

ecological models. We selected a case study of biological control of an invasive aquatic weed (water hyacinth or *Eichhornia crassipes* (Mart.) Solms) in Louisiana due to the rich data sources available, but show that even this data-rich case lacks some critical data for measuring outcomes that are primary program goals.

Our work adds to the existing literature by: (1) providing evidence of the effectiveness of water hyacinth biological control; (2) demonstrating use of spatially and temporally detailed models to estimate ecosystem service benefits of invasive control; and (3) incorporating substitutability of the most valuable ecosystem services to avoid overstating benefits. In addition, we aim to motivate thorough data collection and thereby improve the data landscape for economic studies by highlighting how even small additions to monitoring data can enable economic analysis.

## Case study

In Louisiana, water hyacinth (*E. crassipes* (Mart.) Solms) has been a problem for over a century. Water hyacinth is a widespread invasive floating plant that is often considered one of the world's worst nuisance plants due to its aggressive growth and ability to form large interlocking mats that impede navigation and fishing and impair water quality, among other harms (*Howard & Harley, 1997*; *Center, 2004*; *Villamagna & Murphy, 2010*). Water hyacinth was first introduced into the U.S. in New Orleans, Louisiana in 1884 as an ornamental plant (*Wunderlich, 1962*). Just 15 years later, it had become such an impediment to navigation in the southeastern states that the U.S. Congress authorized the U.S. Army Corps of Engineers (USACE) to clear infestations in navigable waters (*Sanders, Theriot & Perfetti, 1985*).

A primary concern in Louisiana is that water hyacinth disrupts recreational and commercial boating by limiting access and increasing costs of navigation and maintenance. Small to medium size boats with propeller engines aim to navigate around mats since floating mats clog propellers and water-cooling systems and obscure navigational hazards (*South Atlantic Division Field Committee, 1948*; *Obeid, 1975*; *Alimi & Akinyemiju, 1991*). In addition, dense infestations block wharves, piers, and ramps (*Howard & Harley, 1997*). About 80% of Louisiana recreational boaters surveyed in 2009 moderately or strongly supported "improving aquatic weed control" (*Isaacs & Lavergne, 2010*).

A wide array of additional harms has also been documented in Louisiana or similar settings. Despite being a detriment to recreational fishing access, a substantial literature documents both harms and benefits to fish communities from water hyacinth with effects apparently dependent on the fish species present and plant density (*Villamagna & Murphy, 2010*). Water hyacinth impairs waterfowl habitat by reducing open water area and depleting food sources (*Gowanloch, 1944*). Effects on infrastructure include clogged water intakes of power plants, water treatment plants, and other industries (*USAID, 1971*; *Mailu, 2001*; *California Department of Parks and Recreation, Division of Boating and Waterways, 2013*) and reduced drinking water quality (*Mailu, 2001*; *UNEP Global Environmental Alert Service, 2013*). Further, water hyacinth can block streams and

drainage canals, reducing their discharge capacity and increasing flooding risk (*Gowanloch & Bajkov, 1948*).

### History of water hyacinth control in Louisiana

Mechanical removal and herbicide application were the primary treatment strategies of water hyacinth infestations in Louisiana until the early 1970s, when biological control agents were introduced. Biological control, the intentional introduction of a host-specific organism to suppress a target species, has been used as part of the strategy to manage water hyacinth infestations in many southern states and in a number of countries since the 1970s (*Center, 2004*). Four insect biological control agents have been introduced in Louisiana, *Neochetina eichhorniae* and *N. bruchi* from 1974 to 1977, *Niphograpta albiguttalis* from 1979 to 1981, and *Megamelus scutellaris* from 2010 to 2016. *N. eichhorniae*, or the mottled water hyacinth weevil, was the most successful at establishing and dispersing throughout the state (*Coombs et al., 2004*; *Grodowitz, Johnson & Harms, 2014*). *Neochetina* spp. reduce water hyacinth vigor through larval and adult feeding which reduces production, fertility and spread, and increases susceptibility to herbicides (*Goyer & Stark, 1984*; *Gettys et al., 2014*; *Jones et al., 2018*).

A program coordinated between Louisiana Department of Wildlife and Fisheries (LDWF), USACE, and the US Department of Agriculture (USDA) was successful in dispersing *Neochetina* spp. within the state from 1975 to 1977 (*Manning, 1979*). *Neochetina* spp. remain widespread in Louisiana, despite the lack of new releases since the 1980s, although local abundance varies considerably (N. Harms, 2016, personal observation). Variability in weevil populations is likely due to spatially variable habitat conditions including water quality, especially available nutrients, nearby insecticide use and climate. The biological control agent, *M. scutellaris*, was recently released in Louisiana, but colonization and effectiveness at controlling water hyacinth is unknown. No other biological control agents for water hyacinth were being released as of 2017, and herbicide control is the primary active management tool. Approximately 16,600 ha of water hyacinth control were treated annually from 2000 to 2013 (A. Perret, 2015, personal communication).

## METHODS

Using data from multiple government agencies and private businesses, we developed an integrated set of models and analyses to conduct a cost-benefit analysis of invasive water hyacinth control in Louisiana. Of the ecosystem services identified through literature review and interviews with government officials or business owners, four had sufficient data and evidence of responsiveness to water hyacinth cover for quantitative analysis: (1) recreational fishing from boats, (2) recreational waterfowl hunting from boats, (3) boat-dependent businesses (marinas, tourism), and (4) drinking water supply. Other services were identified that could not be evaluated due to lack of data to quantify harms in the absence of invasive control including: commercial fishing, commercial navigation, flood risk mitigation, hydroelectric production, and nonuse values for species.

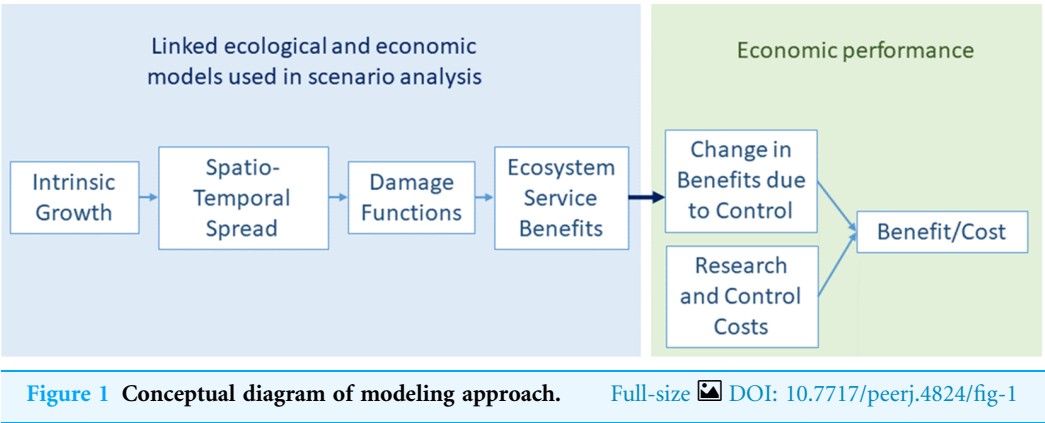

**Figure 1 Conceptual diagram of modeling approach.**

Ecosystem service benefits were measured as the difference between conditions with and without water hyacinth treatment. Four types of models or analyses made up our integrated suite of tools used to estimate biophysical conditions and benefits (Fig. 1): (1) water hyacinth intrinsic growth rate model; (2) water hyacinth spatio-temporal spread model; (3) damage functions relating water hyacinth cover to biophysical changes; and (4) economic benefit transfer analyses. Extensive data were collected to model the with-treatment scenario and historic conditions or data from unmanaged infestations were used to model the counterfactual scenario. Costs were derived from state and federal government records of research and implementation activities for biological control and herbicides. All analyses and models used the time period of 1975–2013 in the state of Louisiana.

## Intrinsic growth rate model (treatment effectiveness)

Water hyacinth distribution data collected in the spring and fall each year (usually April and November) from 1975 to 2013 (*Louisiana Department of Wildlife and Fisheries, 2014*) enabled us to model plant growth rate response to biological control. We fit predicted to observed cover data using a logistic growth model that controlled for winter severity by including a variable of days with minimum temperature at or below freezing (*Nesslage et al., 2016*). The model fit was substantially improved by incorporating a time-varying intrinsic growth rate for water hyacinth such that the growth rate declined through time. Results from this model informed the spatio-temporal spread model, as described below.

## Spatio-temporal spread model

Because many benefits and harms of ecosystem changes depend on where and when people are using the ecosystem, we developed a spatially and temporally detailed model of occurrence by adapting an existing parsimonious model, which had been tested on six invasive species, including water hyacinth (Model D, *Robinet et al., 2012*). The spatio-temporal model included three submodels: (a) habitat suitability model, (b) logistic growth model, and (c) kernel density function to spread plants across the landscape. The first model was used to parameterize spatial differences in growth and carrying capacity and the latter two models were run in sequence, at each time step to evaluate biomass and spread.

We modified the *Robinet et al. (2012)* model by incorporating a time-varying growth parameter in the logistic growth model (to reflect biological control effects), developing a sparse kernel density function adapted to work over large regions, and fit multiple empirical model parameters to Louisiana data (Appendix A, Supplement S1). For example, we estimated water hyacinth carrying capacity per landscape grid cell by fitting ratios of observed levels of maximum cover (*Manning, 1979*) to model-generated estimates of habitat suitability (*Sutherst, Maywald & Kriticos, 2007*, CLIMEX niche maps, D. Kriticos, 2014, personal communication). We also developed kernel density function parameters by calibrating a simulation model to historic data on water hyacinth coverage by major watershed. Cover data from the period prior to widespread biological control release (1975–1978) provided information on conditions prior to biological control that were used to calibrate the model for the counterfactual scenario.

## Damage functions for ecosystem services

We created a general damage function to relate percent cover to loss of benefits and parameterized the function for each ecosystem service. The damage function generates a proportion (unitless value) that is multiplied by the total potential benefits by location and year, in the absence of water hyacinth. The function documents the degree to which boat travel time increases as a function of percent cover of water hyacinth, up to the point at which a waterway becomes impassible.

The damage function was parameterized using findings from landscape ecology since recreational boats navigate around water hyacinth mats in a manner similar to animals navigating a fragmented landscape (*Obeid, 1975*; *Alimi & Akinyemiju, 1991*). Research suggests that the path length that an organism requires to move through a landscape (i.e., connectivity) increases when unsuitable land cover reaches 15–25% of the landscape (*With, Gardner & Turner, 1997*). Further, movement can become impossible when unsuitable cover reaches 70–90% of the landscape (*Mönkkönen & Reunanen, 1999*). Based on these findings, we selected 20% and 80% as the thresholds to represent the water hyacinth cover that would begin to impede and then prevent navigation (Fig. 2).[1] The linear segment of the function was drawn to connect these two points and reflected increasing path length as suitable cover (i.e., open water) was lost (after *With, Gardner & Turner, 1997*). Selected boat operators were consulted by phone and all concurred that our estimates were realistic.

## Economic valuation methods

All ecosystem service benefits were estimated through benefit transfer or costs avoided methods that required multiplying a unit value by the number of affected entities (Table 1). In benefit transfer, the final value was estimated as the total consumer surplus, which is a measure of well-being derived from the difference between what a consumer would have been willing to pay to enjoy a service and what was actually paid (*Rosenberger & Loomis, 2001*; *Freeman, Herriges & Kling, 2014*). Costs avoided were measured as direct expenses avoided.

[1] Some types of boats that use air propulsion rather than propeller engines are able to navigate water hyacinth mats without problems. No data were available to quantify the proportion of such boats in use, however, anecdotal reports suggest that these are not in widespread use for fishing, hunting, and boat tourism.

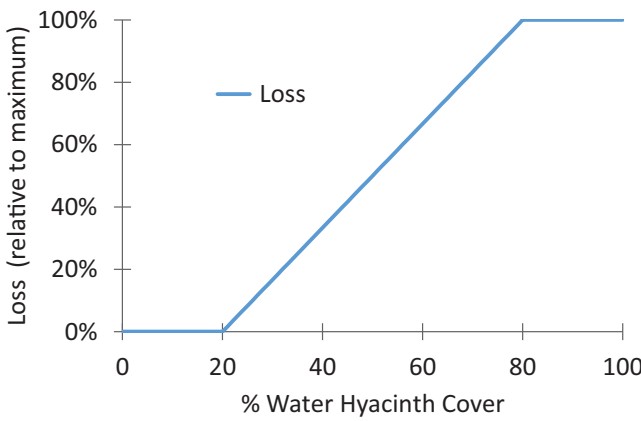

**Figure 2** General damage function used to estimate loss of ecosystem service as a function of water hyacinth density.

**Table 1** Ecosystem services analyzed and associated biophysical and benefit metrics

| Service | Biophysical change (captured in damage functions) | Unit value | Affected entities | Benefit metric |
|---|---|---|---|---|
| Recreational fishing | – Change in boat access | – Consumer surplus per fishing day | – Total user days per season per year (low and high biomass seasons) | – Consumer surplus value of all fishing days adjusted for substitution effects |
| Recreational waterfowl hunting | – Change in boat access | – Consumer surplus per hunting day | – Total user days per season per year (low and high biomass seasons) | – Consumer surplus value of all hunting days |
| Commercial tourism | – Marinas and boat launches blocked | – Average response cost per marina per year | – Number of vulnerable marinas (brackish water only; freshwater unavailable) | – Maintenance costs avoided (mechanically breaking up mats) |
| Drinking water | – Number of water intakes physically blocked by water hyacinth | – Average response cost per treatment plant per year | – Number of water supply treatment plants | – Maintenance costs avoided (mechanically breaking up mats) |

### Benefit transfer methods

Using an existing database (*Rosenberger, 2011*), we estimated a unit value per recreational user day as the average of 19 studies, selected because they represented freshwater fishing or hunting from boats in southern states. We converted values to 2013 dollars using the consumer price index (*Bureau of Labor Statistics (BLS), 2013*). We estimated an average user day consumer surplus of $55.90/day for fishing and $47.46/day for hunting. The hunting consumer surplus was made a function of existing area suitable for hunting per parish, following methods described in *Wainger et al. (2013)*.

Total user days per year were estimated from license sales data (*Louisiana Department of Wildlife and Fisheries, 2015*) and survey data for fishing and hunting. To estimate freshwater fishing days, two surveys were cross-referenced (*Ogunyinka & Lavergne, 2009*; *US Fish and Wildlife Service, 2013*) and only 20% of waterfowl hunting days were used from the national survey (*US Fish and Wildlife Service, 2013*) to represent freshwater activity, as based on a Louisiana survey (*Laborde & Rohwer, 2010*). The ratio of permits

sold to user days was calculated for the year when both data were available and then applied to convert historic data on licenses to total annual user days for all years.

The affected users by location and time step were determined by comparing spatio-temporal model projections of water hyacinth cover with estimated spatial and temporal usage patterns. Recreational fishing days were distributed to each parish using an angler survey (*Ogunyinka & Lavergne, 2009*) and divided into two seasons using monthly crappie fishing effort data, which were the most complete of available data and representative of multiple fisheries (A. Perret, 2016, personal communication). Based on the survey, we estimated that about 65% of fishing effort occurred December–April, when biomass of water hyacinth was estimated to be at low winter–spring levels and 35% occurred the rest of the year when biomass was at or near peak. Hunting days were assigned to parishes based on the percentage of survey respondents that identified a given parish as their favorite (*Laborde & Rohwer, 2010*) and distributed evenly across the months in which hunting was permitted. Waterfowl hunting effort was split 50–50 between low and high biomass seasons.

The final benefit calculation by recreational service, spatial region, and season was the product of user day value and user days, modified by the percent of total benefits per water hyacinth cover, as provided by the damage function (Fig. 2). The benefits per service per year ($V_y$) were a weighted sum of the two seasons, as given by:

$$V_y = \sum_{s=(1,2)} a_s \sum_{r=(1,\ldots,n)} (ud_{yr})(b(c_s)) \tag{1}$$

where $y$ = year; $s$ = season (1 = high biomass, 2 = low biomass); $r$ = region (1, ..., $n$) which represents sub-areas of parishes defined by discrete bins of percent cover; $a_s$ = proportion of annual recreational activity allocated to season; $u$ = value per user day without water hyacinth; $d_{yr}$ = annual user days per year and region. The function $b(c_s)$ is the damage function that provides the proportion of benefits delivered per water hyacinth cover per season ($c_s$). Cumulative benefits were the sum of the 38 years of record (1975–2013).

In the without control scenario, cover becomes dense and widespread in some areas, resulting in most or all fishing days lost. It is likely that under such extreme conditions some anglers would find alternative recreation activities, rather than lose all recreation benefits. A recent survey of freshwater anglers in Australia suggested that 59% of anglers would be willing to substitute a different outdoor recreation, if they could not go fishing (*Sutton & Oh, 2015*). To account for peoples' willingness to substitute another activity for fishing, we assumed a linear increase in substitution between 20% and 80% water hyacinth cover, up to a maximum of 59% substitution. We handled substitution differently for hunting by reducing consumer surplus per additional hunting day as huntable area increased. Additional methods are provided in Supplement S2.

### Avoided cost methods

Costs avoided were judged appropriate to use because businesses that were affected by water hyacinth were expending resources in management. Marina operators respond to presence of water hyacinth by using boats to mechanically break up mats.

The reported costs of control (fuel, labor, and equipment) ranged from $13,000 to $23,000 per marina per year, based on interviews with a small set of operators. Approximately 400 marinas on non-estuarine brackish water (*Louisiana Oil Spill Coordinator's Office, 2004*) were estimated from Geographic Information System (GIS) analysis to be vulnerable to invasion.

Water treatment plant operators, similarly, used boats to mechanically break up water hyacinth mats that clogged intake pipes. The average annual cost of mechanical breakup (including fuel, labor, and equipment) was reported as $2,300 (2013$) or a range of $1,100–$3,800. We identified 77 vulnerable sites using a GIS analysis of surface water intake locations (*Louisiana Department of Health and Hospitals, 2006*).

Avoided costs were scaled to the density of water hyacinth in the vicinity of the marina or treatment plant using the damage function (Fig. 2), parameterized for each service (Supplement S3). For both entities, the damage function was scaled by using the average annual cost at 80–100% cover and assuming spending was zero at 0–20% cover. Cover was assessed in the immediate vicinity (grid cell) of the marina or water treatment plant.

## Cost of water hyacinth research and control

We estimated the total costs of water hyacinth management as the sum of treatment (mechanical, herbicide, and biological control), research and program costs for water hyacinth from 1975 to 2013. Ongoing treatment costs in Louisiana have been borne by the state and the USACE. Research investments were made through the USACE Aquatic Plant Control Research Program (APCRP) on biological, herbicidal and integrated control to manage water hyacinth in the US (1975–1989). US Department of Agriculture, Agricultural Research Service (USDA-ARS) contributed to biological control agent development via efforts such as overseas exploration and host-specificity studies. The USACE Large Scale Operational Management Test (LSOMT) program (in place 1976–1980) primarily funded release and monitoring of biological control agents in Louisiana (*Sanders & Theriot, 1986*). Costs reflected government management activities conducted on public waters and not activities on privately owned lands. Program management costs were included in most data sources but we added a 5% management cost to federal programs when such costs were not provided. Further information is in Supplement S4.

Historical (1975–2013) management costs for Louisiana were consolidated from annual reports and spending data from the LDWF, removal of aquatic growth programs of the USACE New Orleans district (USACE-MVN) and the USACE Mobile district (USACE-SAM), and the USACE Aquatic Plant Control Research Program (USACE-APCRP). When data were unavailable in some years, we estimated missing values by interpolation, so as not to underestimate spending levels. All values were adjusted for inflation (*Bureau of Labor Statistics (BLS), 2013*) and reported in 2013 dollars.

Three different approaches were used to estimate costs of developing biological control agents for water hyacinth. For each of the two initial *Neochetina* spp. agents, we apportioned national program research spending (*Andres, 1977*) to Louisiana using the percentage of water hyacinth habitat contained in that state (23% or 2.3 million ha) relative to all US states, including Hawaii (*Wainger et al., 2016*). Reported costs were not

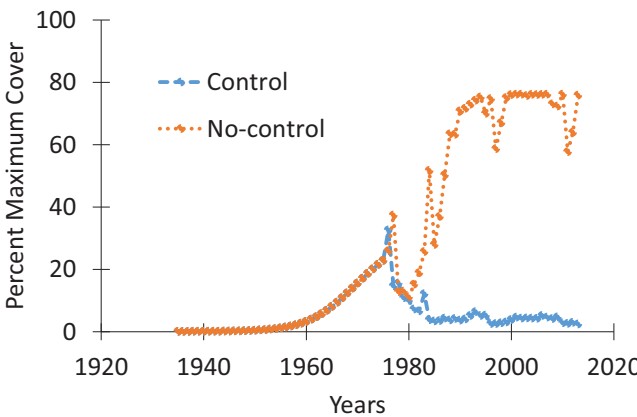

**Figure 3 Simulation of fall water hyacinth density through time, with and without control.**

available for development of two other agents, *N. albiguttalis* or *M. scutellaris*, so we conservatively estimated costs associated with *N. albiguttalis* as the same as *Neochetina* spp., and estimated *M. scutellaris* costs using information provided by principal investigators involved with the project (P. Tipping, 2014, personal communication).

## RESULTS

### Spatio-temporal spread modeling

The results of the spatio-temporal modeling showed substantial differences between the with- and without-control scenarios. The differences in the percentage of water area invaded by water hyacinth were estimated to be 57% higher in the spring and 76% higher in fall 2013 (Fig. 3) without treatment. The dramatic decline in coverage with control appeared to be supported by an 84% decline in growth rates over the study period (1976–2003) (*Nesslage et al., 2016*). The spatial results reflected the north–south gradient in habitat suitability, since abundance was greater in the more suitable (warmer) southern areas (Fig. 4).

Results strongly suggested that biological control had reduced water hyacinth cover because of the observed decrease in growth rate over time and because we found no statistically significant difference in herbicide use between years with high and low extent of water hyacinth (Exact Wilcoxen Rank Test, Wilcox $W = 51$, $p = 0.14$; high extent defined as >400,000 acres statewide), suggesting that herbicide treatment was not responsible for overall historic declines. However, increased efficiency of herbicide treatment may partially explain a lack of correlation between treatment and subsequent declines in cover. We also noted that growth decreased, despite documented increases in average air temperatures over the study period that would tend to increase the growth rate of this tropical plant (*Wilson, Holst & Rees, 2005*).

### Economic benefits

The total value of the four ecosystem service benefits over the 38-year analysis period was estimated as $4.2 billion (Table 2). The vast majority of these benefits were from

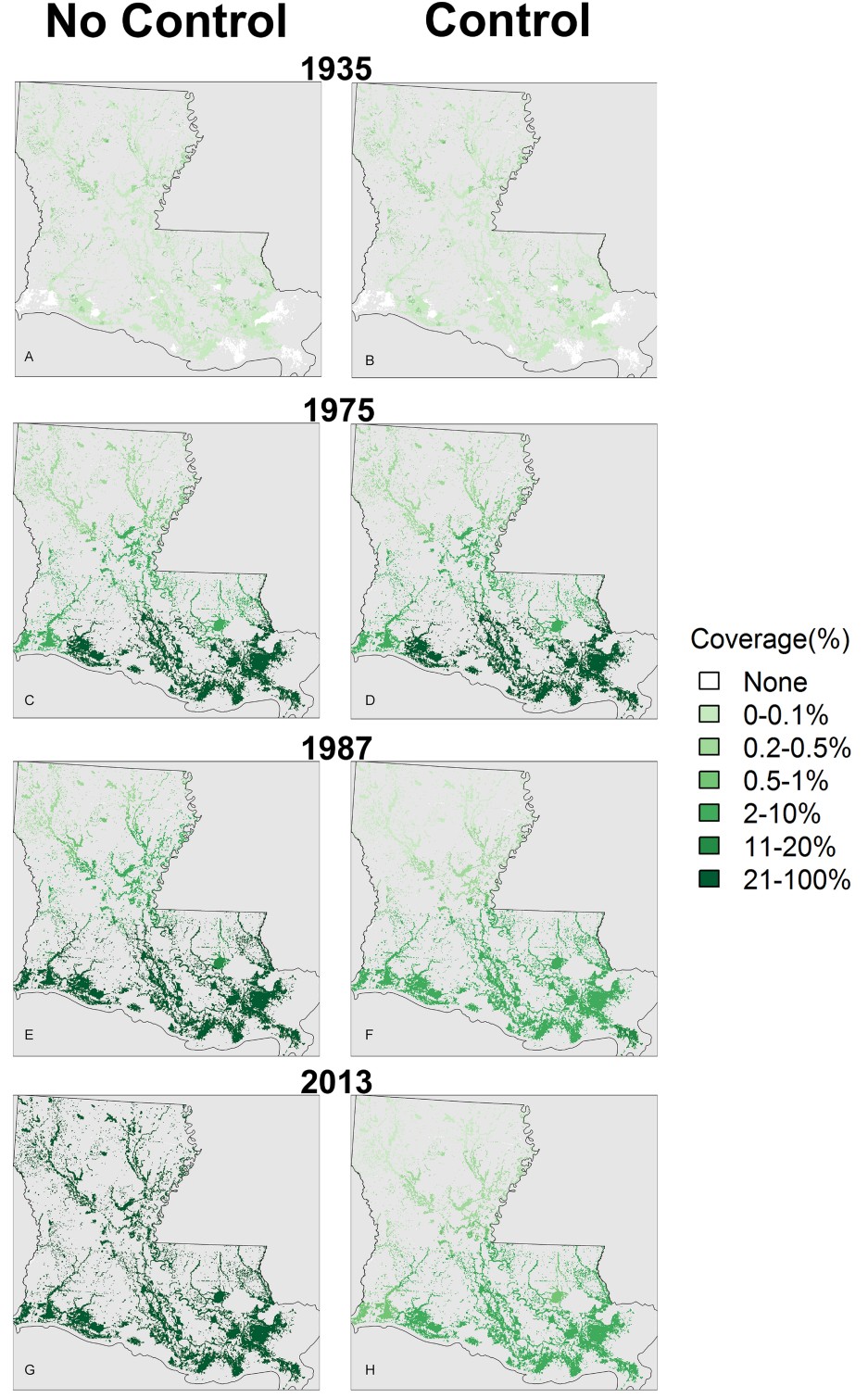

**Figure 4** Water hyacinth coverage per grid cell (as % of water area) for the without (A, C, E, G) and with control (B, D, F, H) scenarios (Fall estimates).

**Table 2 Ecosystem service benefits results (maximum annual and total over 38 years).**

| Ecosystem service | Maximum potential affected users/entities | Annual benefits of control in 2013–final year of analysis (M$2013) | 1975–2013 cumulative benefits (M$2013) |
|---|---|---|---|
| Recreational freshwater fishing (With activity substitution)* | 583,480 anglers | $172 | $3,880 |
| *Recreational freshwater fishing (No activity substitution)* | *583,480 anglers* | *$418* | *$9,450* |
| Recreational waterfowl hunting | 19,400 waterfowl hunters | $7.3 | $164 |
| Commercial boat tourism | 400 marinas (South Louisiana only) | $15.1 | $199 |
| Drinking water supply | 77 drinking water intakes | $0.24 | $3.00 |
| TOTAL (with substitution) | | $612.5 | $4,242 |

Notes:
* Substitution refers to adjustment made to reflect anglers choosing an alternative form of recreation if freshwater fishing is unavailable. See "Benefit Transfer Methods" for details.

**Table 3 Spending on water hyacinth management, research, and development from 1975 to 2013 in Louisiana.**

| Category | Organization | Total program cost (M$2013) | Louisiana cost (23% of research costs, M$2013)* | Time period |
|---|---|---|---|---|
| Herbicide research (APCRP) | USACE-ERDC | $6.74 | $1.55 | 1976–1989 |
| Biological control research (APCRP) | USACE-ERDC | $4.95 | $1.14 | 1975–2014 |
| Biological control development | USDA-ARS | $12.76 | $2.93 | |
| Integrated control research (APCRP) | USACE-ERDC, MVN | $2.98 | $0.69 | 1976–1989 |
| Large Scale Operations Management Test (LSOMT) | USACE-ERDC, MVN | $2.1 | $2.1 | 1975–1980 |
| **Subtotal for research** | | **$29.53** | **$8.41** | 1975–2014 |
| Herbicide application | USACE | | $94.6 | 1975–2013 |
| | LDWF | | $20.7 | 1975–2013 |
| **Subtotal for herbicide application** | | | **$115.3** | 1975–2013 |
| **TOTAL** | | | **$124.36** | |

Notes:
Subtotals and total are shown in bold.
* See Supplement S4 for further details.

preserving recreational freshwater fishing, which would be substantially impacted during times of peak water hyacinth cover. For 2013, the last year of analysis, the annual benefits were $195 million. The correction for activity substitution had a large effect on values, reducing fishing benefits from $418 to $172 million in year 2013. All reported values are 2013 dollars, unless otherwise specified.

## Cost analysis

Costs of research and programmatic expenses of water hyacinth biological control were estimated to total $29.5 million across multiple government agencies or $8.4 million for Louisiana alone (Table 3, subtotals shown in bold). Among the responsible agencies, a total of approximately $115 million was spent on herbicide application of water hyacinth between 1975 and 2013. With that cost added, the total spending for Louisiana was $124 million, reflecting the large proportion of herbicide control costs in management spending.

Missing data create some uncertainty in these values. Some biological control development costs from the 1970s are lacking due to missing data from large scale rearing and large equipment used in releasing agents. Herbicide costs were modestly overestimated (5–10%) because, in some years, program funds were targeted to other plants.

### Cost benefit analysis

When aggregate benefits (Table 2, *Cumulative Benefits*) were divided by aggregate costs (Table 3, *Louisiana Cost*), the benefit cost (B/C) ratio was 34:1, strongly suggesting that the benefits of water hyacinth control well exceeded the program costs. If we adopt the perspective that B/C ratios should be evaluated based on the present value at program initiation, all costs and benefits for our case study should be discounted to a present value for 1975 (*Hill & Greathead, 2000*). When we conducted this analysis, the B/C ratio dropped to 6.8:1 at a 3% discount rate or 2.9:1 at a 7% discount rate. The ratio drops because costs accrue early while substantial benefits accrue many years later. This new B/C ratio based on 1975 present values nonetheless, still suggests that the program is socially efficient, since it more than pays for itself. These analyses show that the B/C ratio is sensitive to perspectives built into the analysis and how future users are reflected in the discount rate (see *Cropper, 2013*).

## DISCUSSION

The high B/C ratio that we found (34:1) suggests that the investment in developing biological control agents was an efficient use of funds. The aggregate costs of $124 million were an order of magnitude smaller than the $4.2 billion in benefits generated in the 38 years of our dataset. This B/C ratio is consistent with results from other programs to control water hyacinth or similar aquatic nuisance species, which have ranged globally from 2.5:1 to 124:1 (representing many undiscounted values) (*McConnachie et al., 2003*; *De Groote et al., 2003*). Ratios for biological control programs across many types of invasive species have ranged from 1 to 1,000:1 (using discounted values) (*Hill & Greathead, 2000*). While many published studies show a substantial B/C ratio, *Hill & Greathead (2000)* suggested that even if some biocontrol had B/C ratios less than one, when viewed as a portfolio of investments, the high returns to some agents makes the risk of development new agents worthwhile.

Although biological control can take time to show effectiveness and does not usually eradicate the target species (*Schooler, McEvoy & Coombs, 2004*), it has potentially long-lasting, self-propagating, and self-sustaining benefits that offset the initial upfront investment costs. As evidence from this program, releases of agents in Louisiana were discontinued in the 1980's, yet overall suppression has been maintained (*Nesslage et al., 2016*). (Recent releases of *M. scutellaris* in Louisiana have not led to confirmed establishment.) In comparison, management of water hyacinth with herbicide requires continual investment.

The regional benefits of control can be hard to perceive given that local populations of water hyacinth remain in many places and may still reach damaging levels. Ongoing

herbicide application, or maintenance control, is needed to keep local populations in check, a result that is not uncommon with aquatic weeds that are controlled with biocontrol. However, our evidence, which was developed at the regional scale, supports the idea that water hyacinth would be a substantially greater problem statewide without biological control. These findings reflect conditions from the peak of the damage from water hyacinth that included spatially extensive and tall mats of water hyacinth, that have not been observed in 50 years or more.

Although measured benefits were substantial, our results likely underestimated benefits since we omitted whole categories of values, particularly nonuse values, that can be comparable to use values (*Navrud, 2001*; *Johnston, Besedin & Wardwell, 2003*) and have been measured for aquatic invasive species cases (*McIntosh, Shogren & Finnoff, 2010*). Further, we were not able to include harms suggested by historic information, including increased flooding risk, damage to infrastructure, and disruptions to commercial navigation (as suggested by *Thunberg, Pearson & Milon, 1992*). These events were not routinely recorded and predated the tenure of most current emergency managers and harbor masters, which limited our ability to collect information needed to construct models for the counterfactual scenario. In addition, we only included harms to marinas on brackish waters but almost twice as many boat trips take place on freshwater systems, compared to all saltwater systems (*Isaacs & Lavergne, 2010*). Further, annual costs to marinas and water treatment facilities were based on relatively modest current infestations and may not be representative of higher infestation levels. Finally, we based our analysis on benefits of agents for only one state, but agents can and are being used in other areas.

These benefit omissions did not alter the fact that we found a high benefit-to-cost ratio, which is one way for objectively evaluating optimal level of control (*Leung et al., 2002*) and ensuring that benefits of management exceed costs and, therefore, generate net social benefits. Without such analyses, decision makers are left to infer appropriate levels of spending based on incomplete, and possibly biased, information. For example, past problems with biological control have caused managers to be risk averse to embarking on biological control, even to the extent that past successes might not have been possible under current decision rules (*Hinz et al., 2014*).

Despite its usefulness in this case, cost-benefit analysis must be used with caution since, frequently, many types of environmental benefits cannot be monetized. Costs can appear to exceed benefits due to data or technical limitations of economic analysis, rather than a lack of net benefit. For example, if the ability of invasive species to alter system resilience to stress is poorly understood, the benefits of preventative actions will be underestimated (*Shackelford et al., 2013*). Non-monetary benefit indicators can be an alternative approach to comparing benefits within a cost-effectiveness framework, when they cannot be monetized (*Wainger & Mazzotta, 2011*).

Our economic study was enabled by thorough record keeping on patterns of human use of the environment. Recreational fishing and hunting effort was tracked through time in databases of licenses sold and separate surveys evaluated spatial usage patterns and

total use. Spatial data on marina and boat launch locations, created in response to the Deepwater Horizon oil spill, enabled the treatment costs avoided analysis.

Time series data or widespread spatial coverage that encompass both invasive species cover and human responses can be used to deepen understanding of tradeoffs associated with different intensities of invasive species treatment effort, relative to cover alone. For example, although we were able to use literature to estimate changes in fishing effort as a function of water hyacinth cover, data that encompassed both fishing effort and cover over a range of conditions would have enabled direct empirical modeling of this effect. Further, data on drainage blockages and upstream water hyacinth cover might have allowed other risks to be estimated to improve evaluation of program success.

Finally, human adaptations to environmental change need to be incorporated in economic analysis to accurately estimate risk. We used survey evidence to suggest how people might make activity substitutions under extreme conditions of a changed environment. However, future work would be strengthened by explicitly documenting any thresholds or non-linearities of human responses with aquatic plant cover, including willingness to make activity substitutions, such as salt-water fishing. All biophysical and socio-economic changes are ideally measured with before/after and control/impact (BACI) design experiments (*Underwood, 1992*), which provide the strongest evidence for isolating treatment effects from natural variability. However, since not all case studies will have thorough studies, results from well-studied locales can be transferred to less studied sites using models and foundational site-specific data.

Although the data in Louisiana were among the most comprehensive that we have seen, our analyses have uncertainty due to some data gaps. We could not cleanly separate effects of biological control and herbicides or demonstrate synergies of the two types of treatment (as suggested by *Van & Center, 1994*; *Center et al., 2002*), because presence and density of control agents were unknown and herbicide data were summarized at the annual, statewide scale. Also, further study of species interactions that result from treatment would clarify whether treating one invasive species can facilitate invasion by another (*Santos et al., 2009*) and undercut economic gains of treatment.

Our study suggested that information does not have to be perfect to conduct economic analyses but that robust analysis is supported by sufficient data and information to demonstrate past management effectiveness or to project future success. Data gaps can be filled with after-the-fact mapping using remote sensing data (*Albright, Moorhouse & McNabb, 2004*) and by using modeling to separate causes from effects. We demonstrated the latter approach by using the spring and fall water hyacinth coverage observations and historic weather records to distinguish effects of overwinter mortality from treatment effects, thereby refuting the hypothesis that cold winters that coincided with biological control release were solely responsible for plant declines (*Nesslage et al., 2016*).

## CONCLUSION

Our cost benefit analysis revealed that water hyacinth control, largely the result of biological control, has generated benefits in Louisiana well in excess of research, development, and implementation costs. Using the four ecosystem services with the

best data, we estimated a B/C ratio of 34:1, due to all forms of control. This high ratio for aquatic plant control is consistent with other literature that safe and effective biological control agents generate net benefits (*Lovell, Stone & Fernandez, 2006*) and can dramatically lower management costs of control (*Hinz et al., 2014*).

An additional implication of this work is that sound economic analysis of ecosystem services relies on a substantial amount of data to quantify the many cause-and-effect relationships that link actions to benefits. Benefit assessments are supported by three fundamental types of observations (1) management effectiveness, (2) ecological outcome changes, and (3) social and economic changes. For natural resource managers, the suggestion is that, if they want to be able to demonstrate economic benefits of programs, they not only track changes in invasive cover (with and without management), but also collect data reflecting potential or observed effects on human activities and use of the environment. For aquatic invasive species, questions to address through data collection include: Does water quality become a problem for aquatic life? Do game animals disappear? Is outdoor recreation participation reduced? Not only must data be collected, but critical data sources must be accessible to researchers over long time periods.

We also suggest that costs of economic analyses can be reduced by using generalizable analytic tools, such as the spatio-temporal spread model on which we built. Complex models that require many years of site-specific research to parameterize may not support cost-effective treatment action (*Simberloff, 2003*). Further, generalizable models may be superior to highly specified models because their projections of future outcomes can be more robust to potential variability (discussed further in *Robinet et al., 2012*).

Our results for water hyacinth in Louisiana are likely to be generalizable to other cases where a safe and effective biological control agent has the potential to suppress an aquatic invasive species that interferes with recreation, boat-dependent businesses, and/or water supply. The high B/C ratio that we found suggests that our estimates are robust to the minor cost data omissions that we noted and would only increase if we had been able to add more ecosystem services. Overall, we find that biological control has the potential to deliver a long-term stream of ecosystems services that can justify initial research investments.

## ACKNOWLEDGEMENTS

We are grateful to Kristen Hychka, Elizabeth Price, Taylor Hollady and Sam Leap of UMCES for assistance with analyses and manuscript preparation. We sincerely appreciate the invaluable data, information, and assistance of the Alex Perret, Jack Isaacs, Michael Harden and others, Louisiana Department of Wildlife and Fisheries; Skip Martin, David Fletcher, Glenn Suir, and Ned Mitchell, USACE; Ted Center, USDA Agricultural Research Service; Pam Fuller, US Geological Survey; Darren Kriticos, CSIRO Australia; Ben Malbrough, Bayou Lafourche Fresh Water District; Ginger Rushing, Assumption Parish Water Plant; Joe Van Marcke, City of Thibodaux; Dirk Barrios, Lafourche Parish Water District No. 1; Michael Lecompte, Consolidated Waterworks District No. 1 of Terrebonne Parish; David Allemond, McGee's Landing; Coerte Voorhies, The Atchafalaya Experience; Allie Cozad, Red River Waterway Commission; Michael Massimi, Barataria-Terrebonne Estuary Program; Theryn Henkel, Lake Pontchartrain Basin Foundation; Chris Tuckey, U.S. Coast

Guard 8th District Waterways Branch; Ted Falgout, Former director of Port Fourchon; and Emily Federer, Port of New Orleans. Thanks also for helpful reviews from Matt Weber, Julie Coetzee and two anonymous reviewers.

### Funding

This work was supported by the US Army Corps of Engineers Aquatic Plant Control Research Program (Cooperative agreement W912HZ-11-2-0046). Al Cofrancesco provided historical context, data and materials necessary to design models to test effects of biological control. He had no role in data analysis nor in the design of the economic analysis. He reviewed but did not change the final manuscript.

### Grant Disclosures

The following grant information was disclosed by the authors:
US Army Corps of Engineers Aquatic Plant Control Research Program: Cooperative agreement W912HZ-11-2-0046.

### Competing Interests

Al Cofrancesco is a director at the US Army Engineer Research and Development Center that funded this work. The remaining authors declare that they have no competing interests.

### Author Contributions

- Lisa A. Wainger conceived and designed the experiments, performed the experiments, analyzed the data, authored or reviewed drafts of the paper, approved the final draft.
- Nathan E. Harms conceived and designed the experiments, performed the experiments, analyzed the data, prepared figures and/or tables, authored or reviewed drafts of the paper, approved the final draft.
- Cedric Magen performed the experiments, analyzed the data, prepared figures and/or tables, authored or reviewed drafts of the paper, approved the final draft.
- Dong Liang performed the experiments, analyzed the data, prepared figures and/or tables, authored or reviewed drafts of the paper, approved the final draft.
- Genevieve M. Nesslage conceived and designed the experiments, performed the experiments, analyzed the data, authored or reviewed drafts of the paper, approved the final draft.
- Anna M. McMurray performed the experiments, analyzed the data, authored or reviewed drafts of the paper, approved the final draft.
- Al F. Cofrancesco conceived and designed the experiments, contributed reagents/materials/analysis tools, approved the final draft.

### Data Availability

S1: Liang, Dong (2018): S1_Water Hyacinth Spatio-Temporal Model. figshare. Fileset. https://doi.org/10.6084/m9.figshare.5904055.v1;

S2: Magen, Cedric; Wainger, Lisa (2018): S2_Benefits to Fishing and Hunting as Treatment Costs Avoided. figshare. Fileset. https://doi.org/10.6084/m9.figshare.5933407.v1;

S3: Magen, Cedric; Wainger, Lisa (2018): S3_Benefits to Boat-Dependent Businesses and Drinking Water Treatment Plants as Treatment Costs Avoided. figshare. Fileset. https://doi.org/10.6084/m9.figshare.5933263.v1;

S4: Harms, Nathan (2018): S4_Estimates of Research and Treatment Costs for Control of Water Hyacinth. figshare. Fileset. https://doi.org/10.6084/m9.figshare.5904637.v2.

## Supplemental Information

Supplemental information for this article can be found online at http://dx.doi.org/10.7717/peerj.4824#supplemental-information.

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
