# Peer review of "Evidence-based economic analysis demonstrates that ecosystem service benefits of water hyacinth management greatly exceed research and control costs"

_PeerJ, doi:10.7717/peerj.4824_

## Round 0.1 · original submission · Minor Revisions

I enjoyed your paper as a very good example of an integrated application of both economic and ecological approaches.
The reviewers and I agree that your manuscript is generally well written and clear. However they provided a few minor suggestions for improvements. For example, rev#2 suggests to slightly re-organize the discussion, starting with a brief overview of the main findings, instead of listing the key weakness of the study.

Reviewer 1 ·

Basic reporting

I found this manuscript easy to read and comprehend. The grammar and sentence structure is well-conceived and presented. Overall and excellent manuscript one that is worthy of publication. The literature review was very good and many of the more important articles were referenced. I did find several cases where materials and methods were repeated in the results section and these need to be located and removed. I do understand that a large part of this manuscript type requires a detailed explanation on logic and processes involved in formulating the approach – maybe it would be prudent to simplify the materials and methods section and include portions in the actual results to help the reader understand the reasoning better. The figures were adequate.

Experimental design

The experimental design was conceived and executed with no real issues. I thought the explanations were very clearly given which allowed me to easily understand the scope and breadth of the research. The authors clearly presented the pros and cons of their approach and adequately justified omissions when needed. While other manuscripts have addressed the issue of the effectiveness of the water hyacinth biocontrol agents the combination of various models adds credence to the generally accepted notion of the agent’s impact.

Validity of the findings

In general I have no real issues with the findings. As stated earlier the authors adequately address strengths and weaknesses of their methods and how they relate to their conclusions. While the impact of the water hyacinth agents have been shown to be effective in previous publications the additional depth the authors went through just adds credence to this accepted idea.
However, to many people the agents are not thought to be effective although numerous manuscripts and popular articles have attempted to validate such. Reasons for this are many but the main one is water hyacinth still exists and in many areas it is still a massive problem. Control methods continue, people still have difficulties navigating through such areas, and there are no real signs that are easily observable to make these people believe the problem is getting better. I had hoped that the authors would address such concerns and attempt to alleviate such issues but I see no evidence of that. Another issue that is not addressed is why these perceptions of less than satisfactory agent’s impact occur. One was addressed above but another is the highly successful use of alligatorweed biocontrol agents – the first target addressed in the aquatic plant world. That program was so inordinately successful that it is recognized as a prime example of weed biocontrol. With the release of the agents on alligatorweed control was achieved in months and not years as is the case for water hyacinth and gave people the false expectation that such amazing control is the norm and not the exception. Perhaps this is outside the realm of this manuscript but might help address some of the perceptions people have as to the non-effectiveness of water hyacinth agents.

Additional comments

I believe I addressed general comments previously. Have the authors re-check citations I did find a few missing though I did not go over the list in detail. Also, a minimal number of more specific comments have been placed in the manuscript itself.

Annotated reviews are not available for download in order to protect the identity of reviewers who chose to remain anonymous.

Reviewer 2 ·

Basic reporting

The article is generally well written, and sufficiently under-built with literature references. I have the following suggestions for improvements:
1. The Introduction section is quite lengthy, and does not read well. I would suggest the authors condense this section by 20-30%, and steer clear of overly-long phrases and rather colloquial language.
2. The Introduction section may benefit from a short additional paragraph in which the (exceptionally-high) benefit:cost ratios of biological control are already illuminated (Greathead & Greathead; Cock et al.), and in which the dearth of ex-post economic studies of this practice is mentioned (evt. citing Naranjo et al., and others).
3. The authors could consider adding in more concrete data in the Abstract - e.g., highlighting the 84% decline in growth rate, $4.45 billion in ecosystem service benefits.
4. The Methods section can be further streamlined. Also, certain parts of the Methods (e.g., line 203-207) may fit better in the Results section.
5. Some reference appear to be missing in the Reference section.
6. There are occasional typos in the text (e.g., line 172, line 481).

Experimental design

Overall, the research questions are properly defined & the methods appear to be sound. I would like to see a greater effort to dis-aggregate control effects due to biological control versus those of 'integrated control' (i.e., combination of mechanical, herbicidal and biological control).
With the wealth of data that the authors have at their disposal, I believe it should be possible to generate spatially-explicit information on amount/coverage of herbicide sprays and mechanical harvesting for a particular year (or multi-year time period). Next, contrasting waterhyacinth cover declines in areas with varying intensity of different control tactics can provide a ways of assessing the level of weed suppression through biological control (as compared to 'integrated control'). These data possibly can also be shown in a graph, or map. Another way of quantifying the relative contribution of biological control is by mapping Neochetina distribution/abundance, and contrasting shifts in usage of alternative measures (mechanical/herbicidal) following the local establishment/impact of a given agent. A third way of contrasting the added benefits of biological control is by contrasting inter- and intra-annual level of cover reduction of waterhyacinth due to mechanical/herbicidal interventions, before/after local establishment of a given agent.

Validity of the findings

The findings are exciting and important, and illuminate the impressive (economic / environmental/ societal) benefits of Eichhornia biological control. Yet, these findings are not emphasized sufficiently in the Discussion and Conclusion sections. I would suggest the following steps to remediate this:
1. Instead of starting off with a listing of key weakness of the study, I'd rather initiate the Discussion section with a brief overview of the main findings, to then go more in depth into each of these in the consecutive paragraphs.
2. The current benefit:cost ratio is high, but does indeed cover 'integrated control'. As also indicated above, I'd appreciate if the authors can make further efforts to split out costs/benefits of biological control versus 'integrated control'. It is possible that the benefit:cost ratio for solely biological control is far higher than for integrated measures (possibly nearing some of the figures as attained in Africa).
3. Though the study centers on Louisiana, it is important to emphasize that the investment in biological control research benefited areas far beyond the state's border (as mentioned briefly in line 347). Hence, spatial coverage of the BC endeavor far surpasses this of a single herbicide spray or mechanical removal of waterhyacinth mats. This should be duly emphasized when discussing benefit:cost results and broader impacts of the biological control campaign.
4. Control costs in Table 3 are eye-opening & clearly show the comparatively low cost of biological control both in terms of research and deployment/application. These kinds of results should receive proper coverage in the Discussion section - and a greater effort could be made to underline the self-propagating nature of biological control, and the sustained accumulation of ES benefits over time (as compared to herbicidal control, which comes with yearly repeating cost) - even while BC releases stopped in the 1980s (line 167).
5. End your Conclusions section with an over-arching statement that -aside from stressing the value of generalizable models- duly illuminates the socio-economic importance of biological control.

·

Basic reporting

This is a very well written manuscriupt that flows logically, with sufficient background to place the study into an ecologically and economically relevant context.

Experimental design

While I am not an economist, the methods were written in such a manner that I understood exactly what the paper set out to do, without having to read up on economic modelling or cost-benefit analyses.
The research highlighted an incredibly important issue: how easy it is to forget how bad a situation used to be, and not value the economic and ecological benefits accrued over a significant period of time. The authors are to be commended on selecting a very pertinent topic to aquatic weed conntrol, with international relevance.

Validity of the findings

The findings of the study highlight the benefit of controlling the world's worst aquatic weed, using an economic cost-benefit analysis. Four benefits were assessed, and used to derive the C:B ratio. The authors then discuss how the value may be underestimated, because other benefits could not be included in the study due to incomplete data, thereby evaluating the relevance of their results.
The C:B ratio derived from this study is sure to be very well cited in the future, in other studies justifying the need for biological control of aquatuic weeds, in general.

Additional comments

I really enjoyed reading this paper, and could find no fault or manner to improve it.

---

## Round 0.2 · accepted · Accept

I think you have fully addressed the remarks and suggestions made by the reviewers and that your changes have further improved the paper.

#